# Diagnosis and management of endometrial hyperplasia: A UK national audit of adherence to national guidance 2012–2020

Ian Henderson[1,2‡], Naomi Black[1,3‡], Hajra Khattak[1,4], UKARCOG Working Group Authors[¶], Janesh K. Gupta[5,6‡], Michael P. Rimmer[1,7‡]*

1 The United Kingdom Audit and Research Collaborative in Obstetrics and Gynaecology, London, United Kingdom, 2 National Perinatal Epidemiology Unit, University of Oxford, Oxford, United Kingdom, 3 Division of Biomedical Sciences, Warwick Medical School, University of Warwick, Coventry, United Kingdom, 4 Elizabeth Garret Anderson Institute for Women's Health, University College London, London, United Kingdom, 5 Institute of Metabolism and Systems Research, University of Birmingham, Birmingham, United Kingdom, 6 Birmingham Women's and Children's NHS Hospital Trust, Birmingham, United Kingdom, 7 Medical Research Council Centre for Reproductive Health, Institute of Regeneration and Repair, University of Edinburgh, United Kingdom

‡ IH and NB share first authorship on this work. JG and MR are joint senior authors on this work.
¶ Membership of UKARCOG Working Group Authors is provided in the Supporting information file (S1 Text).
* Michael.Rimmer@ed.ac.uk

**Data Availability Statement:** We are unable to share the data publicly because use of the data is

## Abstract

### Background

Endometrial hyperplasia (EH) is a precursor lesion for endometrial cancer (EC), the commonest gynaecological malignancy in high-income countries. EH is a proliferation of glandular tissue, classified as either non-atypical endometrial hyperplasia (NEH) or, if the cytological features are abnormal, atypical endometrial hyperplasia (AEH). The clinical significance of AEH is that patients face both a high risk of having occult EC and a high risk of progression to EC if untreated. Recommendations on the care of women with EH were introduced by United Kingdom–wide guidance (Green-top Guide No. 67, 2016). National adherence to guidance is unknown. We aimed to describe the care of patients with EH; to compare the patterns of care for those with EH with national guidance to identify opportunities for quality improvement; and to compare patterns of care prior to and following the introduction of national guidance to understand its impact.

### Methods and findings

In this UK-wide patient-level clinical audit, we included 3,307 women who received a new histological diagnosis of EH through a gynaecology service between 1 January 2012 and 30 June 2020. We described first-line management, management at 2 years, and surgical characteristics prior to and following national guidance for EH using proportions and 95% confidence intervals (CIs) and compared process measures between time periods using multilevel Poisson regression. Of the 3,307 patients, 1,570 had NEH and 1,511 had AEH between 2012 and 2019. An additional 85 patients had NEH and 141 had AEH during 2020. Prior to national guidance, 9% (95% CI [6%, 15%]) received no initial treatment for NEH compared with 3% (95% CI [1%, 5%]) post-guidance; 31% (95% CI [26%, 36%]) and 48%

restricted to quality improvement purposes as per the local registrations made at participating hospitals and in accordance with national guidance in the UK from the Health Research Authority. The data were made available for the purpose of peer review at https://osf.io/mvf3d/. For access to the restricted data set for the purpose of quality improvement, please contact the third party data handler: ukarcog.enquiry@gmail.com.

**Funding:** IH is supported by a MRC Clinical Research Training Fellowship, project no: MR/X006115/1. HK is supported by a NIHR Clinical Lectureship. MPR is supported by a MRC Centre for Reproductive Health research grant, no: MR/N022556/1. The funders had no role in study design, data collection and analysis, decision to publish, or preparation of the manuscript.

**Competing interests:** JG contributed to Green-top Guideline No.67 on behalf of the British Society for Gynaecological Endoscopy.

**Abbreviations:** ACOG, American College of Obstetricians and Gynecologists; AEH, atypical endometrial hyperplasia; BMI, body mass index; BSGE, British Society for Gynaecological Endoscopy; BSO, bilateral salpingo-oophorectomy; CI, confidence interval; COVID-19, Coronavirus Disease 2019; EC, endometrial cancer; EH, endometrial hyperplasia; GTG, Green-top Guideline; HRT, hormone replacement therapy; NEH, non-atypical endometrial hyperplasia; PCOS, polycystic ovarian syndrome; RR, rate ratio; SOGC, Society of Obstetricians and Gynaecologists of Canada; UKARCOG, UK Audit and Research Collaborative in Obstetrics and Gynaecology; WHO, World Health Organisation.

(95% CI [43% 53%]) received an intrauterine progestogen, respectively, in the same periods. The predominant management of women with AEH did not differ, with 68% (95% CI [61%, 74%]) and 67% (95 CI [63%, 71%]) receiving first-line hysterectomy, respectively. By 2 years, follow-up to histological regression without hysterectomy increased from 38% (95% CI [33%, 43%]) to 52% (95% CI [47%, 58%]) for those with NEH (rate ratio (RR) 1.38, 95% CI [1.18, 1.63] $p < 0.001$). We observed an increase in the use of total laparoscopic hysterectomy among those with AEH (RR 1.26, 95% CI [1.04, 1.52]). In the later period, 37% (95% CI [29%, 44%]) of women initially diagnosed with AEH who underwent a first-line hysterectomy, received an upgraded diagnosis of EC. Study limitations included retrospective data collection from routine clinical documentation and the inability to comprehensively understand the shared decision-making process where care differed from guidance.

## Conclusions

The care of patients with EH has changed in accordance with national guidance. More women received first-line medical management of NEH and were followed up to histological regression. The follow-up of those with AEH who do not undergo hysterectomy must be improved, given their very high risk of coexistent cancer and high risk of developing cancer.

## Author summary

### Why was this study done?

- New national guidance was introduced in the United Kingdom with recommendations for the care and surveillance of people with endometrial hyperplasia (EH).

- Comparing patterns of care with these recommendations has identified opportunities for improvement.

### What did the researchers do and find?

- After the guidance, medical treatment of non-atypical hyperplasia increased and more patients achieved histological regression, avoiding hysterectomy.

- Surveillance of hyperplasia for those who do not undergo hysterectomy could be improved.

- A greater proportion of women with atypia diagnosed in 2020 commenced medical management and fewer underwent hysterectomy; the impact of the pandemic on care must be considered as a contributory factor towards this.

### What do these findings mean?

- This work has identified where the care of patients with EH diverged from recommended guidance.

- Clinicians may use these findings to review their local care pathways and quality assurance processes so that they can improve the care of women with EH.

- The main limitation was the retrospective collection of data from routine clinical documentation.

## Introduction

Endometrial cancer (EC) is the commonest gynaecological malignancy of high-income countries and the fourth commonest female cancer in the United Kingdom [1]. The incidence of EC is increasing globally [2], likely driven by obesity and its role in the "unopposed oestrogen hypothesis" [3,4]. EC is preceded by a disordered proliferation of the glandular endometrium termed "endometrial hyperplasia" (EH). EH is divided into a precursor lesion without atypical cytological features ("non-atypical endometrial hyperplasia" (NEH)) and a premalignant condition with atypia ("atypical endometrial hyperplasia" (AEH)). The diagnosis of atypia is based on cellular features such as abnormal nuclear morphology [5]. Both precursor lesions are important to identify and treat because of the risk of progression to EC [6]. NEH has a lower risk of progression of below 5% over 20 years, whereas the risk is higher for AEH, at 28% over 20 years [7]. As well as the risk of progression, AEH may coexist with occult EC in one-third of cases [8]. Previously, both the presence of atypia and architectural complexity were involved in the classification of EH, which led to a higher rate of hysterectomy for pathology with low risk of progression to cancer and undertreatment of endometrial atypia with progestogens [9] In 2014, the revised World Health Organisation (WHO) criteria simplified the criteria to NEH and AEH [10] based on atypia alone.

In the UK, the Royal College of Obstetricians and Gynaecologists and the British Society for Gynaecological Endoscopy (BSGE) introduced a guideline on the management of EH in 2016, the Green-top Guideline No. 67 (GTG) [11]. Prior to this, no national guidance for EH existed, resulting in variation in treatment [12,13]. One study of 281 women found 26% of those with NEH underwent a hysterectomy as first-line management [12]. Conversely, 15% of gynaecologists reported recommending progestogen treatment for the first-line management of AEH [13]. Intrauterine progestogen was only recognised as an option for first-line treatment of NEH following randomised evidence from the past decade [14]. This new GTG recommended classification using the WHO 2014 classification system [10]. The GTG recommended the management of risk factors and/or medical management with a continuous progestogen among women with NEH, reserving first-line hysterectomy, and its risks, for those with AEH or NEH following failed medical management. New recommendations were also made on appropriate follow-up with 2 subsequent biopsies at 6-month or 3-month intervals for women with either NEH or AEH who do not undergo first-line hysterectomy, respectively [11]. New guidance is disseminated to all RCOG members alongside its publication on the RCOG website [15].

The rationale for this national audit by the UK Audit and Research Collaborative in Obstetrics and Gynaecology (UKARCOG) was that the care of women with EH had not previously been evaluated nationally and that introduction of the GTG had introduced new standards for care. We therefore sought to describe the care of patients with EH, compare care with the recommendations of the GTG, and evaluate the impact of the GTG by comparing the pattern of care prior to and following its introduction, testing the null hypothesis that there was no

change in care between these periods. By describing the pattern of care for women with EH, we can identify opportunities for quality improvement that make their care safer.

## Methods

This study is reported as per the Strengthening the Reporting of Observational Studies in Epidemiology (STROBE) guideline (S1 Checklist).

### Population

We included 3,307 women who attended a gynaecology service in a UK hospital and who received a diagnosis of EH on their first endometrial biopsy between 1 January 2012 and 31 December 2020. Hospitals from which data were collected are detailed in S1 Table. We excluded women who did not have data on their first line of treatment following biopsy. We excluded women from 2-year follow-up measures if they transferred their care, died following first-line management, or if 2 years from their initial biopsy had not elapsed.

### Study design

This study was a national audit based on retrospectively collected patient-level data. Clinicians at each gynaecology unit in the UK were approached by UKARCOG regional coordinators and invited to undertake the audit based on a hub-and-spoke model [16]. In the units that responded, the audit was registered and approved by the audit department at each site individually by the local clinician affiliated with UKARCOG. Once approved, local data collectors were advised to consult their local audit department or gynaecology department to identify patients diagnosed with EH between 1 January 2012 and 30 June 2020. This time period was chosen to accord with guidance on the retention of medical records and to capture practice prior to and following GTG introduction. Data were then collected from the primary medical records by the audit team member who was a qualified doctor. The audit team member reviewed the primary records of each patient, including available histology reports, clinical letters, imaging reports, and operation notes.The local team member generated a novel identification number for each patient. The data were submitted via a secure platform to a central database held on a secure server. Once centralised, a second data minimisation process was conducted in which identifiable units codes were converted into novel numerical codes prior to use. Ethical approval was not required for this audit in accordance with UK national guidance on the audit of healthcare data for the purpose of clinical audit and service evaluation [17].

### Outcomes

Outcomes were based on the recommendations of the GTG and on the need to understand its impact on clinical practice. We compared first-line management before and after the GTG. This was classified as:

1. No management (no treatment and no surveillance initiated)

2. Further investigation planned (no treatment plan documented within the first 42 days)

3. Medical management (treatment with a continuous progestogen)

4. Endometrial ablation (not recommended)

5. Hysterectomy

We also compared the provision of weight loss advice, which was not mutually exclusive with other categories. We considered all treatment initiated in relation to the first biopsy or a subsequent biopsy within 42 days of the first to represent first-line management; for example, if the initial plan was for hysteroscopy and within 42 days a hysteroscopy was performed and intrauterine progestogen system inserted, then we considered the first-line management to be the intrauterine progestogen. We determined a 42-day threshold allowed for time to process and report both urgent and routine histological samples and for a clinician to action the result. If a patient commenced medical management while waiting for hysterectomy, then we considered hysterectomy to be the first-line management.

Among women who underwent hysterectomy, we compared the approach (abdominal, laparoscopic, laparoscopic assisted, vaginal, unspecified), and extent (total, subtotal) as well as the completion of salpingo-oophorectomy (salpingo-oophorectomy completed, not completed), including among postmenopausal women with AEH. We compared first-line surgical histology over time to understand whether changes in practice impacted the presence of occult malignancy.

We compared the follow-up schedules for women who did not undergo hysterectomy according to the recommended follow-up schedule ($2 \times 6$-monthly for NEH, or $2 \times 3$-monthly for AEH). When calculating the proportion of women who had an appropriate follow-up schedule, we allowed a biopsy/ follow-up interval of <125 days for AEH or <215 days for NEH; that is, we allowed 1-month flexibility. In order to relate variation in the care of women with EH to outcomes of treatment, we compared regression and hysterectomy over the first 2 years from diagnosis, pre-guidance and post-guidance. We compared follow-up patterns (followed up to resolution by either regression or hysterectomy, follow-up commenced but resolution not identified, no follow-up received) according to histology and time period. We selected a 2-year time period for this follow-up measure to capture the subsequent definitive outcome for those who trialled conservative or medical management in the first instance to then receive follow-up biopsy and hysterectomy if indicated. We confirmed that this was an appropriate time period by checking that the large majority of women had either received no follow-up or had achieved resolution or undergone hysterectomy during this time.

## Exposures

The time of first investigation in secondary care (2012–2015, 2016–2019) was the main exposure of interest. We compared outcomes within disease types (NEH, AEH), which were identified by review of the histology reports. We considered any biopsy results within 42 days of the first biopsy to represent the initial histology; that is, we "upgraded" NEH to AEH if identified on a new biopsy within this time period as this reflected a clinical or histological indication to investigate further before commencing "first-line treatment," including where both blind and hysteroscopic biopsies were obtained prior to the results of the blind biopsy being known.

Data were collected on age (<40, 40 to 49, 50 to 59, 60 to 69, ≥70 years); body mass index (BMI; <25, 25 to 29, 30 to 39, ≥40); a history of diabetes or insulin resistance (diabetes or insulin resistance, none); polycystic ovarian syndrome (PCOS); hypertension (yes, no); hormone replacement therapy (HRT) use (ever-used, never-used); smoking status (current smoker, smoking cessation >6 months previous, never smoked); tamoxifen use (ever-used, never-used); and parity (0, 1, 2, ≥3). Additionally, we defined "postmenopausal" as a presenting complaint of postmenopausal bleeding or age over 60 years and without a presenting complaint that indicated a premenopausal status. Data on these exposures were collected from the medical records, which were reflective of the patient-reported history or clinical measurement in the case of BMI.

## Statistical analyses

The baseline characteristics of women were described using frequencies and proportions. We described the first-line treatment of women, the pattern of follow-up at 2 years, and surgical characteristics using proportions and 95% confidence intervals (CIs) based on clustered standard errors to account for the clustering of women within hospitals. We used multilevel Poisson regression to estimate rate ratios (RRs) with 95% CIs for process measures, comparing post-guidance care with a pre-guidance baseline. We similarly modelled first-line management and 2-year follow-up status over time (year of first biopsy). Additionally, we described the characteristics of women who were diagnosed with NEH or AEH during 2020 and described their first-line treatment. We estimated RRs with 95% CIs, comparing care in 2020 with a post-guidance baseline.

To understand why women with AEH may not undergo hysterectomy, we used multilevel Poisson regression to model first-line hysterectomy on patient characteristics among those with AEH in an analysis of complete cases, both univariably and then multivariably. In the multivariable model, we included all potential explanatory risk factors on the basis that these were known to the clinician and patient and may have informed decision-making. We tested interaction terms between risk factors and time period, comparing predicted probabilities between models with and without interaction terms. In an exploratory analysis, to understand whether the chance of resolution could be improved, we also modelled 2-year histological resolution on mode of first-line medical management (intrauterine, oral, combination) among women with NEH, adjusted for age, BMI, parity, and subfertility, which may affect the selection of route. All statistical analyses were conducted using Stata version 18 (Stata Corp; College Station, Texas).

We made a post hoc modification to our analysis by limiting the time period for first-line treatment in the main analyses to 31 December 2019, after we identified a change in first-line treatment in 2020, coinciding with the Coronavirus Disease 2019 (COVID-19) pandemic. We described the first-line treatment of women diagnosed in 2020 separately in an exploratory analysis. Women who were diagnosed after June 2019 were ineligible for our 2-year follow-up measure, so this measure was unaffected. Additional post hoc modifications to our analysis plan included the test for interactions between risk-factors and time period to explore whether the risk-benefit evaluation of hysterectomy among women with AEH changed following the GTG and the exploratory analysis of 2-year outcome according to route of initial medical management.

## Results

We identified 3,377 women who had a new histological diagnosis of EH between 1 January 2012 and 30 June 2020. We excluded 69 women who had missing data on first-line treatment and 1 woman who died prior to first-line treatment. We included the remaining 3,307 women across 76 hospitals. Of these, 1,655 were diagnosed prior to, and 1,652 were diagnosed following introduction of the national guidance at the beginning of 2016. The study flow diagram is found in Fig 1. Women in the post-guidance group had a higher prevalence of PCOS and a higher proportion of HRT use, whereas a lower proportion had used tamoxifen (Table 1). Other characteristics were similar between groups. In both groups, the commonest decade of life for diagnosis was the sixth, and the commonest WHO BMI category was morbid obesity (BMI >40). The population diagnosed during 2020 is described in S2 Table.

Of the 3,081 included women diagnosed up to 2019, 696 (23%) had NEH and 668 (22%) had suspected AEH prior to the national guidance, and 874 (28%) and 843 (27%) had NEH and AEH, respectively, following the introduction of national guidance. In the 2012–2015

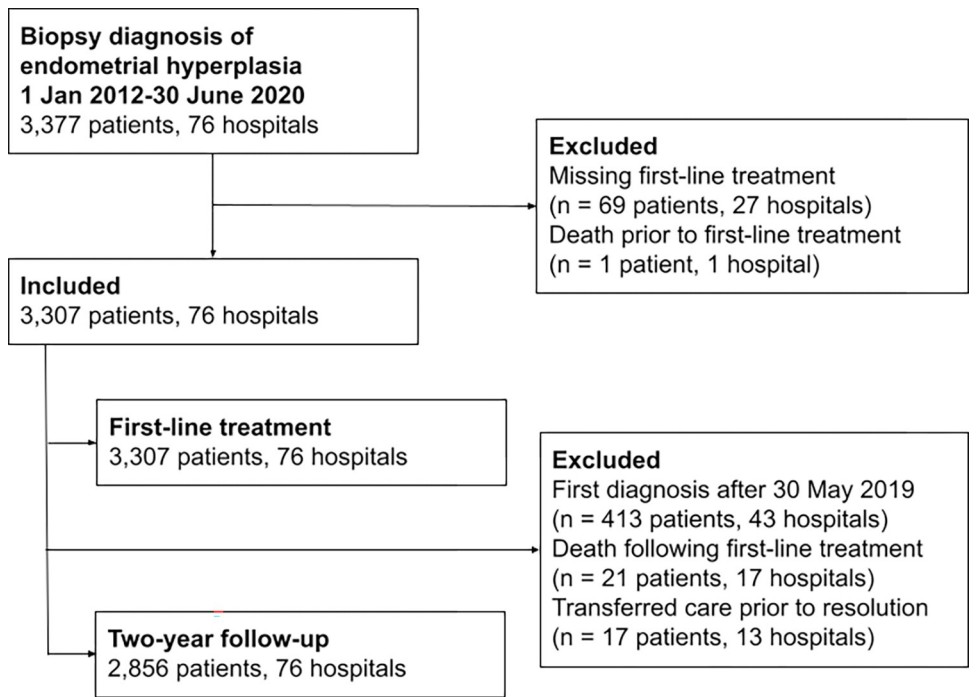

**Fig 1. Study flow diagram of inclusion and exclusion criteria.** Diagram of inclusion and exclusion of patients with endometrial hyperplasia for measures relating to first-line treatment and those relating to 2-year follow-up status.

("pre-guidance") group, the majority of women with NEH (386/696, 55%; 95% CI [48, 62%]) received first-line medical treatment and the majority with AEH (453/668, 68%; 95% CI [61, 74%]) received first-line hysterectomy (Table 2 and Fig 2). In the 2016–2020 ("post-guidance") group, the proportion of women with NEH who received first-line medical treatment increased (594/874, 68%; 95% CI [63, 72%]), whereas the proportion of women with AEH who received first-line hysterectomy remained similar (569/843, 67%; 95% CI [63, 71%]). In particular, the proportion of women with NEH who received intrauterine progestogen increased after the introduction of national guidance, from 31% (214/696; 95% CI [26, 36%]) in the pre-guidance group to 48% (417/874: 95% CI [43, 53%]) in the post-guidance group. Post-guidance, the risk of receiving no first-line treatment decreased (RR 0.36; 95% CI [0.22, 0.59] $p < 0.001$), whereas treatment with first-line intrauterine progestogen increased (RR 1.52; 95% CI [1.28, 1.80] $p < 0.001$) for women with NEH. Additionally, among the 85 women with NEH in 2020 from the onset of the COVID-19 pandemic, 73% (62/85; 95% CI [55, 85%]) received a continuous progestogen and 13% (11/85; 95% CI [6.6, 24%]) received a hysterectomy. Among the 141 women with AEH in 2020, 58% (59/141; 95% CI [46, 69%]) received a continuous progestogen, an increase from the 2016–2019 group (RR 1.62; 95% CI [1.18, 2.21]), $p = 0.003$), whereas 52% (74/141 [95% CI 42, 63%]) received a hysterectomy, a decrease from the 2016–2019 group (RR 0.78; 95% CI [0.61, 0.99], $p = 0.042$). First-line treatment in 2020 is shown in S3 Table.

Among those who did not undergo first-line hysterectomy, a greater proportion were under 40 years of age, had a BMI greater than 40, had diabetes, PCOS, were nulliparous, had a presenting complaint of abnormal uterine bleeding other than postmenopausal bleeding, and had subfertility (S4 Table). Women with AEH who were under 40 years of age were less likely to undergo first-line hysterectomy (aRR 0.23; 95% CI [0.12, 0.43] $p < 0.001$) after adjustment, compared to women 50 to 59 years of age. Women with a BMI greater than 40 were less likely to undergo first-line hysterectomy compared to women with a BMI under 25 in both the

**Table 1. Baseline characteristics before and after guidance.**

| | NEH | | | | AEH | | | |
|---|---|---|---|---|---|---|---|---|
| | Pre-guidance | | Post-guidance | | Pre-guidance | | Post-guidance | |
| | 696 | | 874 | | 668 | | 843 | |
| | N | % | N | % | N | % | N | % |
| Age, mean years (SD) | 54 (12) | | 53 (12) | | 58 (12) | | 57 (13) | |
| Missing | 6 | 0.9 | 6 | 0.7 | 12 | 1.8 | 4 | 0.5 |
| **BMI, kg/m²** | | | | | | | | |
| <25 | 80 | 11 | 98 | 11 | 60 | 9.0 | 94 | 11 |
| 25–29 | 108 | 16 | 146 | 17 | 83 | 13 | 105 | 12 |
| 30–34 | 103 | 15 | 134 | 15 | 88 | 13 | 145 | 17 |
| 35–39 | 86 | 12 | 98 | 11 | 92 | 14 | 131 | 16 |
| ≥40 | 152 | 22 | 212 | 24 | 143 | 22 | 209 | 25 |
| Missing | 167 | 24 | 186 | 21 | 202 | 30 | 159 | 19 |
| **Diabetes** | 94 | 14 | 115 | 13 | 121 | 18 | 138 | 16 |
| **PCOS** | 21 | 3.0 | 57 | 6.5 | 20 | 3.0 | 29 | 3.4 |
| **Hypertension** | 214 | 31 | 231 | 26 | 246 | 37 | 313 | 37 |
| **Smoking** | | | | | | | | |
| Never smoked | 487 | 70 | 570 | 65 | 395 | 59 | 577 | 68 |
| Ex-smoker | 44 | 6.3 | 59 | 6.8 | 39 | 5.8 | 52 | 6.2 |
| Current/recently stopped | 43 | 6.0 | 65 | 7.4 | 48 | 7.2 | 57 | 6.8 |
| Missing | 122 | 18 | 180 | 21 | 186 | 28 | 157 | 19 |
| **Any HRT use** | 40 | 5.8 | 61 | 7.0 | 38 | 5.7 | 44 | 5.2 |
| **Any tamoxifen use** | 50 | 7.2 | 48 | 5.5 | 24 | 3.6 | 40 | 4.7 |
| **Previous births** | | | | | | | | |
| 0 | 113 | 16 | 166 | 19 | 132 | 20 | 165 | 20 |
| 1 | 96 | 14 | 119 | 14 | 67 | 10 | 122 | 14 |
| 2 | 193 | 28 | 258 | 30 | 149 | 22 | 239 | 28 |
| ≥3 | 158 | 23 | 178 | 20 | 119 | 18 | 167 | 20 |
| Missing | 136 | 20 | 153 | 18 | 201 | 30 | 150 | 18 |
| **Presenting complaint** | | | | | | | | |
| Postmenopausal bleeding | 354 | 51 | 451 | 52 | 442 | 66 | 551 | 65 |
| Heavy menstrual bleeding | 189 | 27 | 246 | 28 | 88 | 13 | 123 | 15 |
| Intermenstrual bleeding | 64 | 9.2 | 76 | 8.7 | 34 | 5.1 | 70 | 8.3 |
| Incidental finding | 31 | 4.5 | 43 | 4.9 | 38 | 5.7 | 58 | 6.9 |
| Subfertility | 9 | 1.3 | 10 | 1.1 | 13 | 2.0 | 10 | 1.2 |
| Postcoital bleeding | 17 | 2.4 | 19 | 2.2 | 4 | 0.60 | 18 | 2.1 |

AEH, atypical endometrial hyperplasia; BMI, body mass index; HRT, hormone replacement therapy; NEH; non-atypical endometrial hyperplasia; PCOS, polycystic ovary syndrome; SD, standard deviation.

Proportions may not sum to 100% due to rounding.

univariable (RR 0.74; 95% CI [0.58, 0.94] $p$ = 0.014) and multivariable (RR 0.76; 95% CI [0.57, 1.03] $p$ = 0.075) models, although the strength of evidence in the multivariable model was weak. The associations between risk-factors and first-line hysterectomy among women with AEH are shown in S4 Table and in S1 Fig.

We identified 1,240 women who underwent a hysterectomy for first-line management in both pre- and post-guidance groups. The commonest surgical approach for women who had suspected NEH in the pre-guidance group was abdominal (40/108, 37%; 95% CI [27, 49%]),

**Table 2. First-line treatment.**

| | Time period | | | | | |
| --- | --- | --- | --- | --- | --- | --- |
| | Pre-guidance | | Post-guidance | | | |
| | N | % (95% CI) | N | % (95% CI) | RR (95% CI) | *p*-value |
| **NEH** | 696 | | 874 | | | |
| **First-line treatment** | | | | | | |
| None offered or declined | 65 | 9.3 (5.7–15) | 27 | 3.1 (1.0–5.0) | 0.36 (0.22–0.59) | <0.001 |
| Weight loss (any) | 22 | 3.3 (1.7–6.1) | 52 | 6.0 (3.5–10) | 2.08 (1.23–3.52) | 0.006 |
| Further investigation >42 days | 111 | 16 (13–20) | 106 | 12 (9.1–16) | 0.73 (0.55–0.97) | 0.029 |
| Any continuous progestogen | 386 | 55 (48–62) | 594 | 68 (63–72) | 1.24 (1.09–1.41) | 0.001 |
| IU progestogen | 214 | 31 (26–36) | 417 | 48 (43–53) | 1.52 (1.28–1.80) | <0.001 |
| Oral progestogen | 183 | 26 (19–34) | 182 | 21 (17–25) | 0.86 (0.69–1.07) | 0.18 |
| Endometrial ablation | 13 | 1.9 (0.79–4.3) | 11 | 1.3 (0.53–3.0) | - | - |
| Hysterectomy | 108 | 15 (12–20) | 110 | 12 (9.7–16) | 0.74 (0.56–0.99) | 0.039 |
| **AEH** | 668 | | 843 | | | |
| **First-line treatment** | | | | | | |
| None offered or declined | 4 | 0.61 (0.17–2.1) | 14 | 1.7 (0.92–3.1) | 2.53 (0.81–7.9) | 0.11 |
| Weight loss (any) | 9 | 1.2 (0.49–3.0) | 25 | 3.0 (1.6–5.5) | 2.38 (1.07–5.31) | 0.034 |
| Further investigation >42 days | 41 | 5.9 (3.9–8.9) | 46 | 5.4 (3.7–7.9) | 0.85 (0.55–1.31) | 0.45 |
| Any continuous progestogen | 163 | 24 (17–33) | 213 | 25 (22–29) | 1.10 (0.89–1.36) | 0.37 |
| IU progestogen | 118 | 17 (11–27) | 168 | 20 (17–24) | 1.26 (0.98–1.61) | 0.068 |
| Oral progestogen | 49 | 7.4 (4.9–11) | 51 | 6.0 (4.3–8.2) | 0.79 (0.53–1.18) | 0.25 |
| Endometrial ablation | 1 | 0.15 (0.019–1.2) | 1 | 0.12 (0.016–0.89) | - | - |
| Hysterectomy | 453 | 68 (61–74) | 569 | 67 (63–71) | 0.99 (0.88–1.12) | 0.92 |

AEH, atypical endometrial hyperplasia; CI, confidence interval; HRT, hormone replacement therapy; IU, intrauterine; NEH, non-atypical endometrial hyperplasia; PCOS, polycystic ovary syndrome; RR, rate ratio.

Proportions may not sum to 100% due to rounding.

whereas the commonest approach in the post-guidance group was laparoscopic (57/110, 52%; 95% CI [38, 65%]) (Fig 3). The commonest approach for women who had suspected AEH was laparoscopic in both time periods, with 45% (206/453; 95% CI [34, 57%]) pre-guidance and 56% (319/569; 95% CI [46, 66%]) post-guidance. When considering surgical approach by year, there was an increase in the use of the abdominal and a decrease in the use of laparoscopic approaches in 2020. The majority of women in all groups underwent bilateral salpingo-oophorectomy (BSO) and none in the later period underwent a subtotal hysterectomy. Among women with suspected AEH who were also postmenopausal and who proceeded to hysterectomy, we did not observe a change in the performance of BSO over time; 92% (359/389; 95% CI [89, 94%]) in the early period and 92% (442/ 483; 95% CI [87, 94%]) in the later period.

We identified 26 women who were treated with endometrial ablation in the first instance. Of these 26 women, 25 (25/26, 96%) had a presenting complaint of heavy menstrual bleeding and 1 (1/26, 4%) postmenopausal bleeding; additionally, free-text comments identified that at least 8 women had an ablation at the time of their initial biopsy on which EH was subsequently diagnosed, although this information was not requested; 1 woman who had an ablation had a subsequent hysterectomy.

Pre-guidance, 8 (9.4%; 95% CI [4.5, 18%]) of 85 women with NEH had occult malignancy, whereas in the post-guidance group, 3 (3.5%; 95% CI [1.1, 10%]) of 86 had occult malignancy (Table 3). Pre-guidance, 166 (43%; 95% [34, 53%]) of 364 women with AEH had occult malignancy, whereas in the post-guidance group, 171 (37%; 95% CI [29, 44%]) of 467 had occult

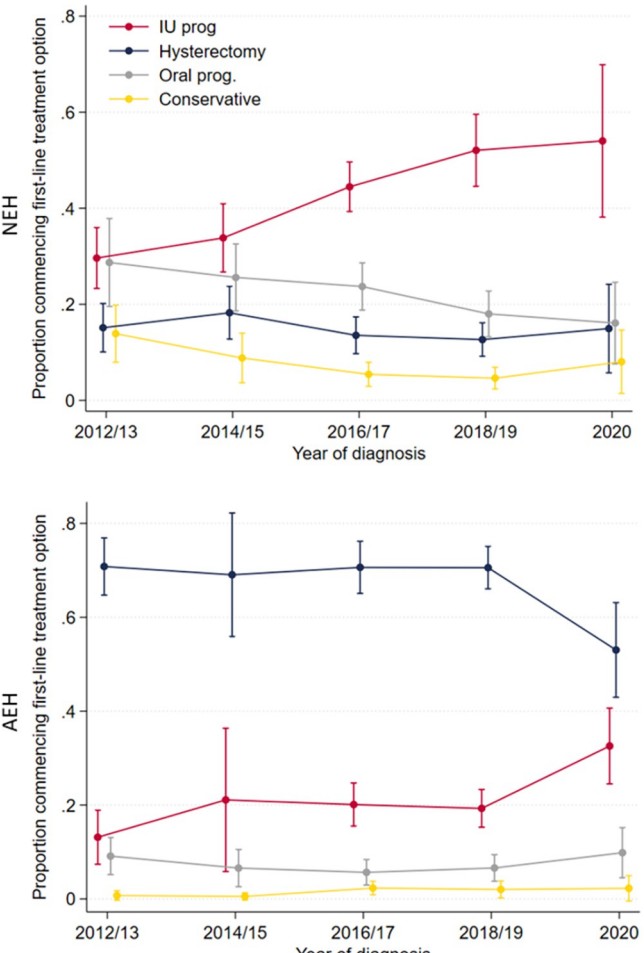

**Fig 2. First-line treatment over time for patients with NEH and AEH.** The proportion of women with non-atypical endometrial hyperplasia (NEH) or atypical endometrial hyperplasia (AEH) treated with intrauterine progestogen("IU prog"), hysterectomy, oral progestogen ("oral prog"), or treated conservatively.

malignancy (Table 3). More than half of the women (52%; 95% CI [42, 62%]) who had an initial diagnosis of AEH who were over 70 years of age were found to have malignancy at their first-line hysterectomy, although the risk was very common at any age. The full characteristics of first-line hysterectomy and surgical histological findings are shown in Table 3.

Follow-up status at 2 years following diagnosis was available for 2,856 women (Table 4). Two years had not yet lapsed for 413 women, meaning that they were ineligible for the 2-year measures. We excluded 21 women who died without a known progression to EC and 17 women who transferred their care prior to definitive treatment. Among women with NEH who did not undergo hysterectomy within 2 years, adherence to an initial recommended follow-up of 2 × 6-month biopsies was 17% (71/415; 95% CI [14, 21%]) pre-guidance and 27% (164/617; 95% CI [22, 32%]) post-guidance. Over the 2-year follow-up period, the commonest follow-up status for patients with NEH in either time period was histological disease regression. The proportion of women followed up to disease regression increased over time, from 38% (264/691; 95% CI [33, 43%]) to 52% (409/789; 95% CI [47, 58%]). The proportion of women with NEH who received no follow-up at all was 21% (145/691; 95% CI [16, 28%]) pre-guidance and 12% (96/789; 95% CI [9.2, 17%]) post-guidance, a decrease (RR 0.65; 95% CI

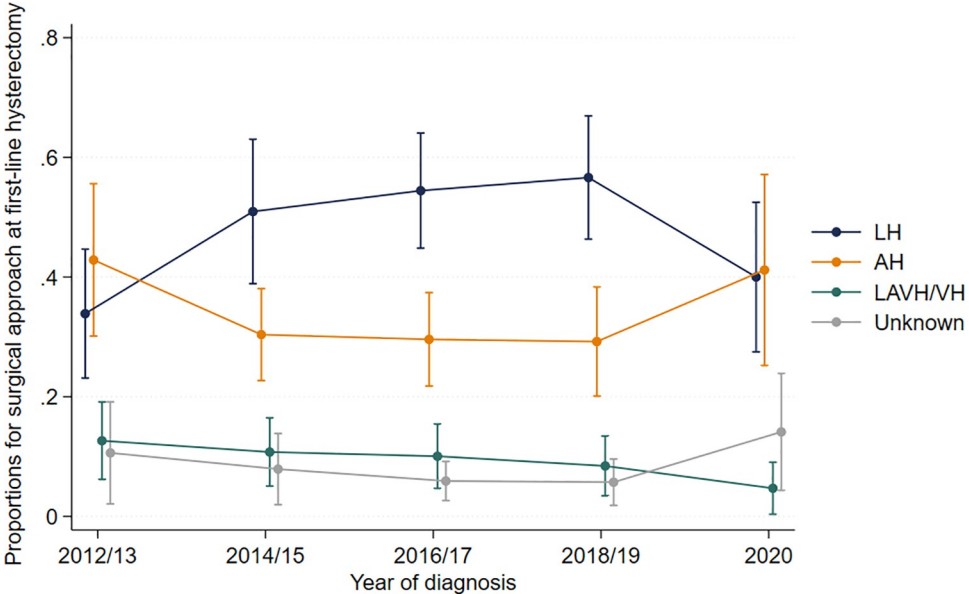

**Fig 3. Surgical approach to first-line hysterectomy over time.** The proportion of patients who underwent laparoscopic hysterectomy (LH), abdominal hysterectomy (AH), either laparoscopically assisted vaginal or vaginal hysterectomy (LAVH/VH) as well as "unknown" type, over time.

[0.49, 0.86], $p$ = 0.003). The proportions for women with AEH were 8.6% (57/660; 95% CI [2.6, 25%] pre-guidance and 3.2% (23/725; 95% CI [1.7, 6.1%]) post-guidance, which also decreased (RR 0.56; 95% CI [0.34, 0.93], $p$ = 0.025). The proportion of women with AEH who received the recommended 2 × 3-month follow-ups was 13% (19/148; 95% CI [8.4, 19%]) pre-guidance and 19% (41/219; 95% CI [14, 26%]) post-guidance. When we grouped women by 2-year intervals for time of diagnosis, the proportion of women with AEH who underwent hysterectomy or who achieved histological regression without hysterectomy remained stable over time, whereas among women with NEH, the proportion who achieved regression increased and the proportion who underwent hysterectomy decreased (Fig 4). The number of women with either NEH or AEH who were followed up to regression over 2 years increased (RR 1.38; 95% CI [1.18, 1.63] $p$ < 0.001) and (RR 1.38; 95% CI [1.00, 1.90] $p$ = 0.047), respectively. Women with NEH followed up to hysterectomy over 2 years decreased (RR 0.72; 95% CI [0.58, 0.90] $p$ = 0.003), whereas for women with AEH, there was no difference (RR 1.01; 95% CI [0.89, 1.14] $p$ = 0.92). We did not observe a difference in the rate of histological resolution at 2 years among women who had first-line medical management for NEH according to the route of progestogen delivery either unadjusted or following adjustment for age, BMI, parity, and subfertility.

## Discussion

We found evidence that introduction of GTG No. 67 was associated with a change in the care of women with EH. Women with NEH were more likely to receive treatment with an intra-uterine progestogen and achieve follow-up to initial histological regression at 2 years and less likely to undergo hysterectomy both as a first-line treatment or within 2 years of diagnosis. There was no difference in the proportion of women with AEH who underwent hysterectomy, which was commoner among all women with EH prior to introduction of the guidance. The quality of follow-up appeared to improve post-guidance; in particular, the proportion of women with NEH who did not receive any follow-up decreased. Nevertheless, there is a need

**Table 3. Surgical characteristics of first-line hysterectomy according to suspected disease type.**

| | Time period | | | | | |
|---|---|---|---|---|---|---|
| | Pre-guidance | | Post-guidance | | | |
| | N | % (95% CI) | N | % (95% CI) | RR (95% CI) | *p*-value |
| **NEH** | 108 | | 110 | | | |
| Approach | | | | | | |
| Abdominal | 40 | 37 (27–49) | 31 | 29 (20–40) | 0.77 (0.48–1.24) | 0.29 |
| Laparoscopic | 38 | 35 (23–50) | 57 | 52 (38–65) | 1.42 (0.92–2.19) | 0.11 |
| Lap-assisted | 14 | 13 (5.6–27) | 7 | 6.5 (2.8–14) | 0.57 (0.21–1.53) | 0.26 |
| Vaginal | 4 | 3.7 (1.3–10) | 4 | 2.8 (0.85–8.7) | 0.82 (0.17–3.92) | 0.80 |
| Unspecified | 12 | 11 (4.7–24) | 11 | 10 (4.0–24) | - | - |
| Total hysterectomy | 108 | - | 108 | - | - | - |
| BSO | 83 | 77 (66–85) | 85 | 78 (67–86) | 1.01 (0.75–1.37) | 0.94 |
| **Surgical histology** | | | | | | |
| Benign finding | 41 | 48 (36–61) | 35 | 41 (29–55) | 0.83 (0.53–1.31) | 0.43 |
| NEH | 24 | 28 (19–40) | 35 | 40 (28–54) | 1.45 (0.85–2.48) | 0.17 |
| AEH | 12 | 14 (7.7–25) | 13 | 15 (10–25) | 1.08 (0.49–2.37) | 0.84 |
| Cancer | 8 | 9.4 (4.5–18) | 3 | 3.5 (1.1–10) | 0.38 (0.099–1.41) | 0.15 |
| Missing | 23 | - | 24 | | - | |
| **AEH** | 453 | | 569 | | | |
| Approach | | | | | | |
| Abdominal | 161 | 36 (27–44) | 168 | 30 (22–38) | 0.83 (0.65–1.05) | 0.12 |
| Laparoscopic | 206 | 45 (34–57) | 319 | 56 (46–66) | 1.26 (1.04–1.52) | 0.016 |
| Lap-assisted | 40 | 8.8 (5.2–15) | 52 | 8.9 (5.2–15) | 0.82 (0.51–1.31) | 0.4 |
| Vaginal | 7 | 1.5 (0.56–4.2) | 2 | 0.36 (0.086–1.5) | 0.22 (0.045–1.11) | 0.066 |
| Unspecified | 39 | 8.6 (3.6–19) | 28 | 5.0 (2.8–8.6) | - | - |
| Total hysterectomy | 449 | 99 (98–100) | 569 | - | - | - |
| BSO | 411 | 91 (87–93) | 509 | 90 (86–92) | 0.99 (0.87–1.12) | 0.84 |
| **Surgical histology** | | | | | | |
| Benign finding | 42 | 11 (7.4–16) | 40 | 8.6 (6.1–12) | 0.82 (0.52–1.30) | 0.40 |
| NEH | 31 | 8.1 (5.6–12) | 36 | 7.8 (5.2–12) | 1.01 (0.62–1.66) | 0.96 |
| AEH | 145 | 38 (30–46) | 220 | 47 (39–56) | 1.21 (0.97–1.51) | 0.097 |
| Cancer | 166 | 43 (34–53) | 171 | 37 (29–44) | 0.86 (0.69–1.08) | 0.19 |
| Missing | 69 | | 102 | | | |

AEH, atypical endometrial hyperplasia; BSO, bilateral salpingo-oophorectomy; CI, confidence interval; NEH, non-atypical endometrial hyperplasia; RR, rate ratio.

Excluding 3 women who had either a clinical or a radiological suspected malignancy despite histological findings.

Proportions for surgical histology results do not include women with missing data.

for the follow-up of women with either NEH or AEH to improve as only a minority received the recommended follow-up post-guidance, despite the well-characterised risk of malignancy among both groups. Many women still underwent an abdominal hysterectomy post-guidance. We observed that more women with AEH diagnosed in 2020 received first-line medical management. This change coincided with disruption from the COVID-19 pandemic. Given that between a third and a half of these women had occult cancer, clinicians must ensure these women diagnosed from 2020 onwards were appropriately followed up and that care has returned to the pre-pandemic standard.

One of the key recommendations of the guidance was on first-line medical management with a continuous progestogen for women with NEH. Intrauterine progestogen, in particular,

**Table 4. Follow-up status at 2 years from diagnosis.**

| | Time period | | | | | |
|---|---|---|---|---|---|---|
| | Pre-guidance | | Post-guidance | | | |
| | N | % (95% CI) | N | % (95% CI) | RR (95% CI) | |
| **NEH** | 691 | | 780 | | | |
| **Followed up to resolution** | **469** | **68 (61–74)** | **591** | **76 (72–79)** | **1.12 (0.99–1.26)** | **0.076** |
| Followed up to initial regression | 264 | 38 (33–43) | 409 | 52 (47–58) | 1.38 (1.18–1.63) | <0.001 |
| Followed up to hysterectomy | 205 | 29 (25–34) | 182 | 23 (19–28) | 0.72 (0.58–0.90) | 0.003 |
| **No follow-up received** | **145** | **21 (16–28)** | **96** | **12 (9.2–17)** | **0.65 (0.49–0.86)** | **0.003** |
| No follow-up, discharged | 77 | 11 (7.7–16) | 58 | 7.4 (5.1–11) | - | |
| Planned follow-up did not occur | 9 | 1.3 (0.064–2.6) | 17 | 2.2 (1.3–3.5) | - | |
| No follow-up, unknown reason | 59 | 8.8 (4.8–15.6) | 21 | 2.8 (1.5–5.1) | - | |
| **Follow-up commenced** | **63** | **9.1 (7.3–11)** | **64** | **8.1 (6.3–10)** | **0.89 (0.63–1.27)** | **0.53** |
| Followed up, ongoing | 1 | 0.14 (0.019–1.1) | 13 | 1.7 (0.72–3.8) | - | |
| Discharged before resolution | 18 | 2.6 (1.6–4.2) | 18 | 2.3 (1.5–3.6) | - | |
| Planned further follow-up did not occur | 25 | 3.6 (2.4–5.4) | 24 | 3.4 (2.5–4.8) | - | |
| Followed up discontinued, unknown reason | 19 | 3.2 (1.8–5.4) | 9 | 1.3 (0.58–2.8) | - | |
| Did not attend | 14 | 2.0 (1.0–4.0) | 29 | 4.0 (2.8–5.6) | - | |
| Progression to cancer, no hysterectomy | 0 | - | 0 | - | - | |
| **AEH** | 660 | | 725 | | | |
| **Followed up to resolution** | **577** | **87 (74–94)** | **671** | **93 (90–95)** | **1.06 (0.95–1.18)** | **0.33** |
| Followed up to initial regression | 68 | 10 (7.8–13) | 105 | 14 (11–19) | 1.38 (1.00–1.90) | 0.047 |
| Followed up to hysterectomy | 509 | 76 (66–84) | 566 | 77 (73–80) | 1.01 (0.89–1.14) | 0.92 |
| **No follow-up commenced** | **57** | **8.6 (2.6–25)** | **23** | **3.2 (1.7–6.1)** | **0.56 (0.34–0.93)** | **0.025** |
| No follow-up, discharged | 45 | 6.7 (1.5–26) | 9 | 1.2 (0.35–4.3) | - | |
| Planned follow-up did not occur | 4 | 0.60 (0.18–2.0) | 7 | 0.96 (0.49–1.9) | - | |
| No follow-up, unknown reason | 8 | 1.9 (1.1–3.4) | 7 | 2.1 (1.2–3.6) | - | |
| **Follow-up commenced** | **14** | **2.1 (1.2–3.7)** | **22** | **2.9 (1.6–5.2)** | **1.30 (0.65–2.60)** | **0.46** |
| Followed up, ongoing | 1 | 0.15 (0.020–1.1) | 10 | 1.4 (0.54–3.4) | - | |
| Discharged before resolution | 4 | 0.60 (0.23–1.6) | 2 | 0.28 (0.067–1.1) | - | |
| Planned further follow-up did not occur | 3 | 0.60 (0.26–1.4) | 7 | 0.83 (0.33–2.0) | - | |
| Followed up discontinued, unknown reason | 6 | 1.2 (0.50–2.8) | 3 | 0.69 (0.30–1.5) | - | |
| Did not attend | 5 | 0.75 (0.35–1.6) | 7 | 0.96 (0.47–2.0) | - | |
| Progression to cancer, no hysterectomy | 7 | 1.0 (0.40–2.7) | 2 | 0.41 (0.014–1.2) | - | |

AEH, atypical endometrial hyperplasia; CI, confidence interval; NEH, non-atypical endometrial hyperplasia; RR, rate ratio.

may offer benefits over non-intrauterine progestogens [14], including a potentially better response among women with morbid obesity [18]. We found that use of first-line intrauterine progestogens increased and that less women with NEH were untreated in the post-guidance period. There is limited international guidance on NEH for comparison, although the Society of Obstetricians and Gynaecologists of Canada (SOGC) recommends medical management only if conservative management fails [19]. We did not observe any obvious differences in the pattern of first-line treatment of women with AEH, other than an increase in documented weight loss advice and weak evidence of a potential increase in the use of intrauterine progestogens, although the first-line hysterectomy rate remained consistent across the 2012–2019 periods. The American College of Obstetricians and Gynecologists (ACOG) similarly recommends total hysterectomy in their guidance on AEH [20]. When we considered why the decision may be made against hysterectomy for women with AEH, we found that those under 40

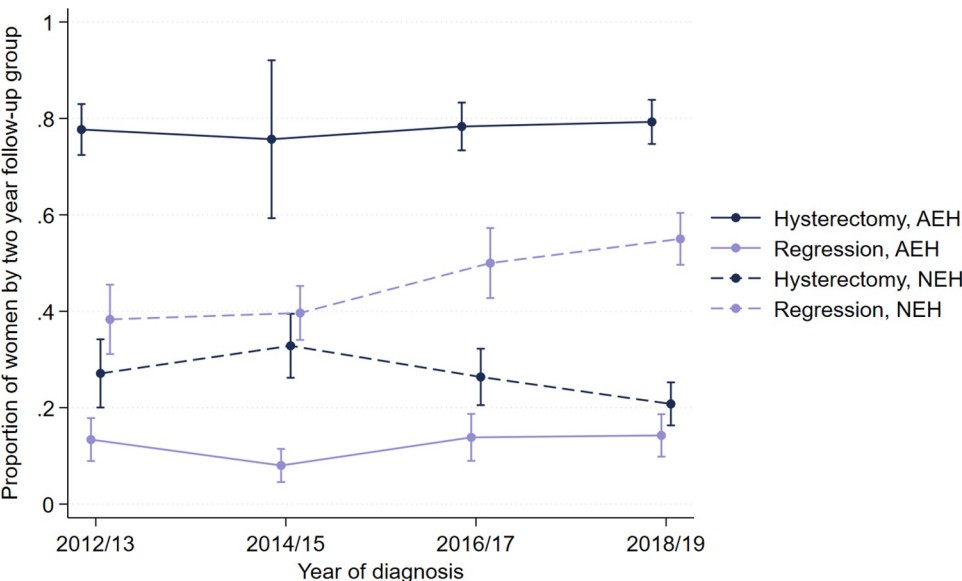

**Fig 4. Proportion of hysterectomy or regression by 2 years over time.** The proportion of patients who were followed up to hysterectomy or histological regression (on at least 1 biopsy) at 2 years from diagnosis over time, according to type of endometrial hyperplasia.

years old (compared with those 50 to 59) or over a BMI of 40 (compared with a BMI under 25) were less likely to undergo first-line hysterectomy. In the former group, this is likely related to fertility wishes; in the latter group, this may be related to either the perceived fitness for surgery or the risk of surgical complication. Obesity confers a greater risk of morbidity in women undergoing hysterectomy with the excess risk greatest for abdominal hysterectomy [21]. We identified an increase in hysterectomies performed laparoscopically, which may reflect the broader move towards laparoscopic surgery and dissemination of these skills over time. Laparoscopic and vaginal approaches offer a lower risk of wound complication and shorter postoperative stay among women with severe or morbid obesity, although there is an approximate 10% rate of conversion to abdominal hysterectomy [22]. Among women with either AEH or early-stage EC, a multicentre Dutch RCT, in which approximately 40% of women were obese, reported no difference in the rate of major complication between total abdominal or total laparoscopic approaches but lower blood loss, use of analgesia, shorter hospital stay, and faster recovery with a laparoscopic approach [23]. Robot-assisted laparoscopic hysterectomy may reduce the rate of conversion to abdominal hysterectomy for women with obesity [24], and the uptake of this approach may benefit women with EH, who have a high prevalence of morbid obesity. Approximately 30% of women with EH still underwent abdominal hysterectomy between 2016 and 2020, greater than the conversion rate. This may mean that some hospitals may not be able to offer all women laparoscopic hysterectomy. Although we could not comprehensively assess why many women did not undergo laparoscopic hysterectomy, the provision of laparoscopic hysterectomy for women with EC differs geographically based on routine administrative data [25]. We did not collect data on additional complicating factors such as previous surgery nor on the size of the surgical specimen given the need to avoid morcellation among women with AEH, which may influence the surgical approach.

A possible explanation for the increase in the proportion of women with AEH who received first-line medical management in 2020 is the impact of the COVID-19 pandemic. Although women with AEH should have been able to access timely first-line hysterectomy given their

high risk of malignancy, there is evidence that the COVID-19 pandemic impacted gynaecological services [26] and provision of cancer surgery [27]. There is evidence that some healthcare professionals offered hormonal treatment and deferred surgical treatment for low-grade EC [28] and, therefore, potentially also AEH. Alternatively, women with AEH may have opted not to proceed to surgical management, given the added risks of hospitalisation and hospital-acquired infection, although most patients wished to proceed with care of gynaecological cancers [29]. Clinicians should ensure their counselling is consistent with pre-pandemic norms in line with guidance and women should be counselled on both the very high risk of concurrent cancer as well as the risk of progression to EC [11]. The medical management of EC is not recommended unless a patient is unfit for surgery [1]. If a woman with AEH decides to proceed with hysterectomy, this should be performed on a cancer pathway by a gynaecological oncologist.

The strengths of this national audit were its large and multicentre population and the detailed level of patient-level data collection. The data were collected by doctors with speciality training in gynaecology, and the use of supplementary free-text comments meant that uncertainties could be described and appropriately coded following centralisation of the data. A review of medical records provided a comprehensive understanding of care and follow-up; nevertheless, we relied on the availability of routine clinical documentation to understand the decision-making process, and some data were missing. We sought to audit cases consecutively, but we cannot be certain that case identification was exhaustive; nevertheless, we do not believe that case retrieval would differ systematically. We could not determine the reason some patients were not followed up if this was not documented. A quarter of patients were missing data on BMI. In our complete case analysis of the association between comorbidity and first-line hysterectomy for AEH, we assumed that in a high-risk clinical setting with decision-making informed by surgical benefit and risk that BMI is less likely to differ systematically. If women with missing BMI did have higher BMI, it is unlikely these would be more likely to have had first-line hysterectomy. We did not include women who died when we considered 2-year follow-up status where there was not a preceding outcome. Women who died were described as having died unrelated to their endometrial disease or from EC, but we cannot exclude that the cause of death was driven by an underlying malignant process. Finally, we audited cases of EH from before and after the introduction of the GTG. We cannot state that the GTG was the only factor underlying any change, and while some recommendations may reflect broader changes in attitude, for example, relating to a laparoscopic approach, we believe it was likely to be a main driver of changes in care during the study period.

The large majority of women with AEH proceeded to hysterectomy or initial histological regression in the 2 years from diagnosis; however, the initial follow-up of women with AEH who did not undergo first-line hysterectomy differed from the recommendation for 2 consecutive 3-month biopsies. Repeat investigation is critical in this group as many of these women will already have an occult malignancy and the decision not to proceed to hysterectomy could be better-informed by information that they did have cancer, if subsequently identified. Clinicians should ensure that evidence-based care is provided as appropriate for the individual patient. All women who elect for medical management of EH should be followed up and those with AEH should be counselled on their high risk of occult cancer. Although early discharge or "did not attend" represented a small minority overall, these are examples of better-characterised reasons for loss of follow-up and may be opportunities to improve the quality of care. Equally, processes for actioning and communicating histological results must be robust. It is critical that women with AEH who did not undergo hysterectomy are followed up with 3-monthly biopsies or else are appropriately counselled so that their decision not to is informed. Local gynaecology units may wish to consider methods to strengthen the follow-up

of women with AEH, including creating a designated lead for patients with ongoing AEH. A local or central EH register may ensure more rigorous patient surveillance and would facilitate further research into the treatment and progression of the condition. General practitioners who may be providing care for women with suspected EH should refer these patients back to their gynaecology service for histological follow-up until safe discharge.

Our findings have identified potential areas for research to improve the quality of care. Interventions to improve the follow-up of women in different situations may be of benefit. Research into patient-centred communication, including patient information leaflets or decision aids, may help to support patients to understand the rationale for proposed treatment and help them decide on their line of treatment. Similarly, patient information leaflets specific to NEH and AEH may help to support the provision of high-quality counselling and health literacy around EH, which may increase follow-up and reduce non-attendance. From a surgical perspective, research on how to improve the dissemination of skills in laparoscopic hysterectomy, including within a very high BMI population, may improve the quality of care. Research into the risks and benefits of robot-assisted hysterectomy among women with obesity for premalignant or early EC may also help to characterise the potential role for this surgical approach in EH given the high rate of obesity in this group.

In this national audit of the management of EH, we found increased uptake of medical management and a decrease in hysterectomy in women without atypia following the introduction of national guidance. While there was some improvement in the quality of follow-up, the majority of women did not receive the recommended surveillance, including for women with premalignant disease. Women with suspected AEH must be appropriately counselled, treated, and followed up, given their very high risk of occult EC.

## Supporting information

**S1 Checklist. Strengthening the reporting of observational studies in epidemiology (STROBE) checklist.**
(DOCX)

**S1 Text. UKARCOG Working Group Authors.**
(DOCX)

**S1 Table. Participating hospitals and their associated National Health Service trust.**
(DOCX)

**S2 Table. Baseline characteristics of patients who were diagnosed with non-atypical or atypical endometrial hyperplasia during 2020.**
(DOCX)

**S3 Table. First-line treatment of patients who were diagnosed with non-atypical or atypical endometrial hyperplasia during 2020 and comparison with a pre-pandemic baseline (2016–2019).**
(DOCX)

**S4 Table. Proportion of patients who underwent first-line hysterectomy and unadjusted and adjusted rate ratios for first-line hysterectomy according to their characteristics.**
(DOCX)

**S1 Fig. Association between risk-factors and first-line hysterectomy for patients with AEH.** Rate ratios with 95% confidence intervals for first-line hysterectomy for the mutually adjusted risk-factors. The baseline group for age is 40–49 years, for BMI is <25, and for parity

is para 2 or greater. Some levels of age, BMI, and parity were combined where these estimates were near identical.

(PDF)

## Author Contributions

**Conceptualization:** Ian Henderson, Naomi Black, Hajra Khattak, Janesh K. Gupta, Michael P. Rimmer.

**Data curation:** Ian Henderson, Naomi Black, Hajra Khattak, Michael P. Rimmer.

**Formal analysis:** Ian Henderson.

**Investigation:** Naomi Black, Michael P. Rimmer.

**Methodology:** Ian Henderson.

**Project administration:** Ian Henderson, Naomi Black, Hajra Khattak, Janesh K. Gupta, Michael P. Rimmer.

**Resources:** Michael P. Rimmer.

**Supervision:** Janesh K. Gupta, Michael P. Rimmer.

**Validation:** Ian Henderson, Naomi Black.

**Visualization:** Ian Henderson.

**Writing – original draft:** Ian Henderson.

**Writing – review & editing:** Ian Henderson, Naomi Black, Hajra Khattak, Janesh K. Gupta, Michael P. Rimmer.

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
