## [Editor Report · Decision Letter 0]

16 Jun 2023

Dear Dr Henderson, 

Thank you for submitting your manuscript entitled "A National Audit of Endometrial Hyperplasia: Comparison of Care with National Guidance" for consideration by PLOS Medicine.

Your manuscript has now been evaluated by the PLOS Medicine editorial staff and I am writing to let you know that we would like to send your submission out for external peer review.

Please re-submit your manuscript within two working days, i.e. by Jun 20 2023 11:59PM.

Kind regards,

Katrien Janin, PhD

Senior Editor

PLOS Medicine

---

## [Decision Letter · Decision Letter 1]

13 Oct 2023

Dear Dr. Henderson,

Thank you very much for submitting your manuscript "A National Audit of Endometrial Hyperplasia: Comparison of Care with National Guidance" (PMEDICINE-D-23-01671R1) for consideration at PLOS Medicine. 

Your paper was evaluated by a senior editor and discussed among all the editors here. It was also sent to independent reviewers, including a statistical reviewer. The reviews are appended at the bottom of this email and any accompanying reviewer attachments can be seen via the link below:

[LINK]

In light of these reviews, we will not be able to accept the manuscript for publication in the journal in its current form, but we like to consider a revised version that addresses the reviewers' and editors' comments. We cannot make any decision about publication until we have seen the revised manuscript and your response, and we plan to seek re-review by one or more of the reviewers. 

We expect to receive your revised manuscript by Nov 03 2023 11:59PM. Please email us (plosmedicine@plos.org) if you have any questions or concerns.

We look forward to receiving your revised manuscript. 

Sincerely,

Katrien Janin, PhD

PLOS Medicine

plosmedicine.org

GENERAL: 

Please organise your manuscript as follows and use the following headings: Abstract, Introduction, Methods, Results, Discussion, Conclusion (1 paragraph)

Please provide 95% CIs and p values for all results were appropriate (including the abstract), check and amend throughout. We suggest reporting statistical information in the following format: ‘x’; (95% CI [‘y’,’ z’] p value) For p values, please report as p<0.001 and where higher as 'p=0.002'. Please add the statistical method used to your method section.

Please define all abbreviations for the reader at first use

For Figure and Tables (including in SI) , please redefine all abbreviations in the legends. 

For in-text reference, citations are placed within square parentheses and should precede punctuation. Please check and amend throughout. 

Please use line numbers for ease of referencing during the revision process.

STUDY DESIGN:

For all observational studies (including audits), in the manuscript text, please indicate: (1) the specific hypotheses you intended to test, (2) the analytical methods by which you planned to test them, (3) the analyses you actually performed, and (4) when reported analyses differ from those that were planned, transparent explanations for differences that affect the reliability of the study's results. If a reported analysis was performed based on an interesting but unanticipated pattern in the data, please be clear that the analysis was data-driven.

Please ensure that the study is reported according to the STROBE guideline, and include the completed STROBE checklist as Supporting Information. Please add the following statement, or similar, to the Methods: "This study is reported as per the Strengthening the Reporting of Observational Studies in Epidemiology (STROBE) guideline (S1 Checklist)."

TITLE: 

Please revise your title according to PLOS Medicine's style. Please note that country and study design should be in the title and please place the study design ("A randomized controlled trial," "A retrospective study," "A modelling study," etc.) in the subtitle (ie, after a colon). 

DATA AVAILABILITY:

The Data Availability Statement (DAS) requires revision. For each data source used in your study:

In your case, it seems option c is applicable, but we require a bit more information (e.g reason for the data restriction and please not that note that a study author cannot be the contact person to request access to the data.

ABSTRACT:

Please structure your abstract using the PLOS Medicine headings (Background, Methods and Findings, Conclusions).

Abstract Background: 

Provide the context of why the study is important. The final sentence should clearly state the study question.

Abstract Methods and Findings:

Please include the study design, population and setting, number of participants, years during which the study took place, length of follow up, and main outcome measures.

Please quantify the main results (with 95% CIs and p values). Suggest reporting statistical information for clarity in the following format: ‘x’; (95% CI [‘y’,’ z’] p value) (e.g Prior to national guidance, 9% (95% CI [6%,15%] p value) received …..). For p values, please report as p<0.001 and where higher as 'p=0.002'.

In the last sentence of the Abstract Methods and Findings section, please describe the main limitation(s) of the study's methodology.

AUTHORS SUMMARY:

In the final bullet point of ‘What Do These Findings Mean?’ Please include the main limitations of the study in non-technical language.

INTRODUCTION:

Please expand your introduction, address past research, context of study, and explain the need for and potential importance of your study. 

METHODS:

Population: please include the number of participants

DISCUSSION:

Please present the Discussion as follows: a short, clear summary of the article's findings; what the study adds to existing research and where and why the results may differ from previous research; strengths and limitations of the study; implications and next steps for research, clinical practice, and/or public policy; followed by a one-paragraph conclusion.

Please consider expanding discussion on global endometrial hyperplasia care in comparison to the UK care. 

ACKNOWLEDGMENTS/ DECLARATIONS

Please remove all statements apart from acknowledgements, author contributions and abbreviations from the main manuscript and include these only in the relevant parts of the manuscript submission form. Funding, competing interest, and data availability will be compiled as metadata.

REFERENCES:

Please use the "Vancouver" style for reference formatting, and see our website for other reference guidelines https://journals.plos.org/plosmedicine/s/submission-guidelines#loc-references

Please ensure that any references to online-only sources include a date of accession (e.g see reference 10) 

Comments from the reviewers:

Reviewer #1: This manuscript is an interesting and ambitious audit of endometrial hyperplasia (EH) treatment over a 9-year period in the UK. The results show a generally high level of guideline-consistent treatment yet identify specific areas where adherence to guidelines could be improved. The methods are strong and the analysis & results are presented in a clear and understandable way. Below are a few questions & suggestions. 

Introduction - Is it possible to summarize what the generally accepted treatment patterns or tendencies were in the UK before the introduction of the GTG in 2016? Or to provide any data on how consistent, or varying, treatment for NEH and AEH were before the GTG was introduced? 

Materials & Methods - on p. 5 the population section doesn't include numbers; later these are provided, in results, but as I read the population section I wished those numbers had been provided there. 

Under Study Design, the collection of data from "physical or electronic records" is mentioned briefly. More details about that step could provide information readers could use to assess the quality of the data collection. 

Also, this section mentions data collection at the level of the individual patient, whereas the Figure 1 refers to "units", which presumably are clinical units. Adding patient-level data to the figure could provide broader context to which patients were included. 

Under outcomes, is there a specific rationale for choosing 42 days as the time interval for first-line management? Is that based on any recommendation or guidance? 

The statistical analysis section mentions relative risks; the model or methods for generating those should be added. Because those metrics appear to be before-vs-after comparisons--i.e., comparing all 2012-2016 patients with all 2017-2020 patients--are those metrics in fact closer to odds ratios, rather than relative risks? That is, the result describes the odds of a given treatment among a clinical subgroup before the GTG vs. after the GTG. 

This comment is strictly a personal preference: the results are thorough and clear to navigate, but the tables are underwhelming. The nature of these comparisons--percentages before vs. after; changes in percent treated before 2016 vs. after--make them well-suited to presentation of data in figures, rather than just tables. 

Discussion

Overall, this section accurately summarizes the results and addresses important contextual factors. Some of the phrases in the first paragraph struck me as perhaps moving beyond what the data showed or could support. Some examples: 

The first sentence includes "were more likely to avoid hysterectomy"; avoid implies hysterectomy would not be justified, but in some cases it might be. 

In the middle of the first sentence, "Nevertheless, there is a need for the follow-up of women with either NEH or AEH to improve" seems incomplete; in other sections of the Discussion, specific areas for improvement are provided, and those sections are much more convincing. 

The final sentence mentions the need for urgent follow-up of patients whose care might have been interrupted by Covid in 2020; at the time of this review, it's already 3 years after those 2020 visits. 

In the Research implications, the text appears to put most of the responsibility for improved follow-up on patients: e.g., patients should receive better communication or pamphlets. The assumption appears to then be that this would empower patients to express their preferences, but this whole section seems to under-appreciate or recognize the role that providers and the clinical care system have in influencing patients' decisions. This speaks to a broader area that is not really covered in the manuscript: the area of implementation science. Some comment on what steps the UK gynecological oncology community made to disseminate the GTG, train providers, and provide continuing monitoring & education for achieving optimal first-line and follow-up care could be useful in this manuscript. 

Reviewer #2: Thanks for inviting me to review this manuscript, assessing the real-world practice for endometrial hyperplasia in UK. A quasi-experimental method was used to assess the possible causality of practice guideline recommendation on the 1st line of treatment. I have the following comments that the investigators may develop further if it improves the study quality.

1 Selection bias. I saw that large number of patients were excluded due to lack of minimum 2-year follow-up (>400 patients). This was to assess the 2-year time point hysterectomy rate in the second step of analysis. The investigator did not provide the data-driven rationale to use 2-year cut-point. 

Majority of EH respond to progestin therapy within 1-year time period. So, among the 400 patients who were excluded due to <2 years of follow-up after the first-line treatment, it is likely that there were many patients already regressed and may not have needed hysterectomy. 

To minimize this selection bias, the investigators might want to consider time-dependent analysis with competing risk. For instance, event of interest (hysterectomy) can be scored as 1 occurred during the follow-up. Then, competing event can be scored as 2 (e.g, death) occurred during the follow-up. Patient who did not have these vent at last follow-up can be scored as 0. Difference-in-Difference (DoD) analysis will be considered to compare pre- and post-guideline implementation periods.

2 Information on temporal trend for NEH / AEH is missing but this will provide important insights. Is AEH increasing over time?

3 Subsequent hysterectomy rates differ between IUD vs systemic progestin. This can be factored in the analysis.

4 Abstract conclusion second sentence. This sentence is unclear and might benefit of clarification. For example, there is no description for occult cancer in abstract result section. 

Reviewer #3: Alex McConnachie, Statistical Review

Henderson et al present the results of a nationwide audit of the management of endometrial hyperplasia in the UK, covering a 9-year period spanning the introduction of new guidance. This review considers the use of statistics in the paper.

Overall, if thought the paper was nicely written, giving clear explanations of where the data came from, and what the exposures and outcomes were. Saying that, I found the paragraph at the top of page 7 to be a little confusing (I think lines 320-326, but the numbering on my copy of the paper is not fully visible). Hopefully this makes sense to a clinical reader. The statistical methods seem reasonable, using basic descriptive methods, accounting for clustering at the hospital level when calculating confidence intervals.

There is no explanation of the sample size for the audit. For an audit, I would not necessarily expect a formal power calculation, but was there a rationale for studying this number of hospitals (are these the only hospitals in the UK that treat these patients, or just the ones that responded?) or for this particular time frame (why 9 years, and not 10)?

The methods section says that relative risks are calculated, and that they "modelled first-line hysterectomy on patient characteristics". A little more information could be given here - what kind of models are we talking about? If the results are presented as relative risks, was it a Poisson model? How did the model account for hospital clustering? Were any covariates included in the models?

Incidentally, when looking at an outcome that is not necessarily an adverse event, such as receiving a particular kind of treatment, I always find the term "relative risk" to be out of place. Is "rate ratio" an acceptable alternative?

The reporting of results in the main text of the paper is fine, except for the tendency to repeat things. For example, the bottom line of page 9 includes "31% (216/ 701, 31% [95% CI 26-36%])" - the "31%" is repeated. In other places, it is the numerator that is repeated, for example on page 11 I saw "25 (25/26, 96%)". None of this is wrong, just a little jarring.

In Table 1, would it be of value to show NEH and AEH separately, pre- and post-guidance? The total columns could be dropped to make space. Also, for age and BMI, were these only recorded in categories, or are the continuous values available? If so, why not report as mean and standard deviation, or as median and quartiles? It might make the differences (if any) easier to see.

In tables 2-4, I think the columns would be better ordered differently. If the main exposure of interest in the introduction of the guidance, then I would have two columns for NEH (pre- and post-guidance) side by side, followed by the same for AEH. That way it is easier to compare the different time periods.

Also in tables 2-4, I would try to include the RR estimates and confidence intervals from Supplementary Table 2. If space is an issue, the p-values could be dropped. I think it is important to see the data presented by time period alongside the estimated associations, to get the full picture.

Supplementary Table 3 looks at first-line hysterectomy in relation to patient characteristics. It shows percentages with different characteristics, according to whether first line management was hysterectomy or not. For me, this is the wrong way round. Here, the outcome is hysterectomy (or not), so rather than showing column percentages (i.e. of those with hysterectomy, how many were in each age group?) it should show row percentages (in each age group, how many had hysterectomy?).

If presented in this way, the no hysterectomy column would not need to be shown. E.g., for age <40, the table could show that there were 130 women with AEH, and 27 (21%) received first line hysterectomy. You could then have a second column showing the RR estimates and confidence intervals for the associations between each patient characteristic and the likelihood of receiving first line hysterectomy. This would complement Figure 5 in the paper. Incidentally, why are diabetes and hypertension not shown in Supp Table 3 (or is "blood glucose disorder" the same as diabetes?).

When looking at these associations, you could look at each patient characteristic individually, or you could try to fit a multivariable model, considering all predictors simultaneously.

Finally, and I may be going to far at this point, the table could also be split into pre- and post-guidance columns. You could then ask whether the factors predictive of first line hysterectomy are the same post-guidance as they were pre-guidance. Ideally, this would be done by fitting regression models with interaction terms, but there is probably not much power in such analyses. Even so, to look at these associations in the two time periods may be interesting, even if the same factors predict hysterectomy in both periods.

[LINK]

---

## [Decision Letter · Decision Letter 2]

15 Dec 2023

Dear Dr. Henderson,

Thank you very much for re-submitting your manuscript "A National Audit of Endometrial Hyperplasia: Comparison of Care with National Guidance" (PMEDICINE-D-23-01671R2) for review by PLOS Medicine.

I have discussed the paper with my colleagues and the academic editor and it was also seen again by the reviewers. On the condition that the remaining editorial and production comments are dealt with, I am pleased to say that we are planning to accept the paper for publication in the journal.

[LINK]

We expect to receive your revised manuscript by the 27th of December. However, we also also conscience that the winter holiday season is upon us. If you like to request an extension, please email us (plosmedicine@plos.org).

We ask every co-author listed on the manuscript to fill in a contributing author statement. If any of the co-authors have not filled in the statement, we will remind them to do so when the paper is revised. If all statements are not completed in a timely fashion this could hold up the re-review process. Should there be a problem getting one of your co-authors to fill in a statement we will be in contact. 

We look forward to receiving the revised manuscript by Dec 27 2023 11:59PM.   

Sincerely,

Katrien Janin, PhD

Senior Editor 

PLOS Medicine

plosmedicine.org

Comments from the Editors:

Thank you for carefully revising your manuscript, and we thank the authors for their responses.

We like to suggest you include “2012-20” somewhere in your title. 

At line 50-51, please add when was the guideline introduced

The abstract should detail the number of events of events for outcomes (as opposed to only percentages).

Also in the abstract, please include the absolute risk(s) of relevant outcomes, not just relative risks or correlation coefficients. (example for absolute risks: PMID: 28399126). 

In the Discussion section, we also ask that you discuss in detail the absolute risk as compared to the relative risk presented regarding the implications and how these findings should be interpreted by the public/individual. We feel that this is extremely important with this type of study, so that readers can appropriately interpret the risk data. Please also provide the absolute risk data as an SI (or where you feel appropriate)

In line with the statistical reviewer we invite you to review the risk reporting and take care to not overinflate interpretable risk.

If you have any question about the above, please do hesitate to contact me directly at kjanin@plos.org

Comments from Reviewers:

Reviewer #1: The authors have been responsive to the reviewers' concerns. Multiple components of the methods and results are clearer and more fully described. 

Reviewer #2: There is one typo-graphic error. Please see page 9 line 274 for clean copy. 

Data were collected on age (<40, 40-49, 50-59, 60-69, ≥40 years);

>=40 will be >=70

Reviewer #3: Alex McConnachie, Statistical Review

I thank the authors for their responses to my previous comments, all of which are satisfactory.

I have just a few very minor corrections to suggest:

- Line 332: should this be "…women with AEH"?

- Line 337: "decreased by 70%" is not accurate, given the RR of 0.36

- Line 363: should this be "…[46, 66%]) post-guidance."?

- Line 405: should be "decreased by 28%"

[LINK]

---

## [Editor Report · Decision Letter 3]

12 Jan 2024

Dear Dr Henderson, 

On behalf of my colleagues and the Academic Editor, Sarah Stock, I am pleased to inform you that we have agreed to publish your manuscript "Diagnosis and Management of Endometrial Hyperplasia: A UK National Audit of Adherence to National Guidance 2012-20" (PMEDICINE-D-23-01671R3) in PLOS Medicine, pending attention to the editorial requests listed below. Please be assured that these are all very minor presentational points/text edits.

Before your manuscript can be formally accepted you will also need to complete some formatting changes, which you will receive in a follow up email. Please be aware that it may take several days for you to receive this email; during this time no action is required by you. Once you have received these formatting requests, please note that your manuscript will not be scheduled for publication until you have made the required changes.

Thank you again for submitting to PLOS Medicine; we look forward to publishing your paper. If you have any questions or any issues arise during the post-accept process, please feel free to reach out directly to me (hvanepps@plos.org) or Katrien (kjanin@plos.org).

Kind regards,

Heather

Heather Van Epps, PhD

Executive Editor

[on behalf of]

Katrien G. Janin, PhD 

Senior Editor 

PLOS Medicine

Editorial requests:

1. Abstract/Background: “EH is a proliferation of glandular tissue, non-atypical endometrial hyperplasia (NEH).” Something appears to be missing from this sentence; please reword for clarity; the distinction between NEH and AEH is much clearer in the Introduction. 

2. Abstract/Background: “…patients face both a high-risk of having current but occult EC and a high risk of progression to EC if untreated.” I’m not sure you need to include ‘current’; suggest rewording to “…patients face both a high risk of having occult EC and a high risk of progression to EC if untreated.” [Note also that the dash in high-risk should be removed].

3. Abstract/Methods and Findings. I would recommend combining the first 2 sentences to avoid an incomplete sentence. Eg, “In this UK-wide patient-level clinical audit, we included 3,307 women…”

4. Abstract/Methods and Findings: please remove semi-colons after percentages throughout the Abstract, and add ‘%’ after both numbers in the 95% CI values throughout. Eg, “…9%; (95% CI [6, 15%]) received no initial treatment…” becomes “…9% (95% CI [6%, 15%]) received no initial treatment…”

5. Abstract/Conclusions: Please abbreviate non-atypical endometrial hyperplasia, as this was defined earlier in the Abstract.

6. Please remove the list of keywords from the manuscript.

7. Introduction, line 183: please change “dissemination” to “disseminated” (“New guidance is dissemination to…”).

8. Introduction, line 188-189: please change the sentence to use patient-centric language; “sought to describe the care of EH…” becomes “…sought to describe the care of patients with EH…”

9. Methods, line 298-299: Please remove the dashing in risk-factors (x2) and decision-making (eg, risk factors, decision making). 

10. Methods, lines 310-312: Please change “A second post-hoc modifications to our analysis plan…” to “Additional post-hoc modifications to our analysis plan included…following the GTG and the exploratory analysis…”

11. Results, line 320-321 (of PDF), please insert a call-out to Table 1 where the data are first mentioned, and remove the standalone sentence referring to the table. Eg, “Women in the post-guidance group had a higher 321 prevalence of PCOS and a higher proportion of HRT use whereas a lower proportion had used tamoxifen (Table 1).” Delete the sentence, “The population is described in Table 1.”

12. Results, lines 330-331: “Of the 3,307 included women, 696 had NEH and 668 had suspected AEH prior to the national guidance, and 874 and 843 had NEH and AEH, respectively, following the introduction of national guidance and up to 2019.” As this sentence refers specifically to those diagnosed up to 2019, it should be modified as follows for clarity (percentages should also be added), “Of the 3,081 included women diagnosed up to 2019, 696 (23%) had NEH and 668 (22%) had suspected AEH prior to the national guidance, and 874 (28%) and 843 (27%) had NEH and AEH, respectively, following the introduction of national guidance.” This also avoids the appearance of missing data (as the numbers did not total 3,307). 

13. Results, line 334: Please include a call-out to the display items where the first pertinent data are mentioned; ie, “…majority with AEH (453/668, 68%; 95% CI [61, 74%]) received first-line hysterectomy (Table 2, Figure 2).” You can then delete the sentence at line 339: “The findings for first-line management are found in Table 2; first-line treatment over time is shown in Figure 2.”

14. Results, lines 336-339: Please reword the sentence for clarity. For example, “In particular, the proportion of women with NEH who received intrauterine progesterone increased after the introduction of national guidance, from 31% (214/696; 95% CI [26, 36%]) in the pre-guidance group to 48% (417/874; 95% CI [43, 53%]) in the post-guidance group.” (or similar).

15. 

Results, lines 340-342 (and throughout): We ask that you do not describe changes as a ‘percent increase/decrease’; rather, you should simply state the data and uncertainty intervals. Eg, “Post guidance, the risk of receiving no first-line treatment decreased (RR 0.36; 95% CI [0.22, 0.59] p<0.001) whereas treatment with first-line intrauterine progesterone increased (RR 1.52; 95% CI [1.28, 1.80] p<0.001) for women with NEH.” There are many examples of this throughout the Results section.

16. Results, line 359: please include a call-out to Suppl. table 3 at the end of the sentence ending “…other than postmenopausal bleeding, and had subfertility (Table 3).” You can then delete the sentence at line 356-357: “The characteristics of women with suspected AEH who did not receive first-line hysterectomy are described in Supplementary Table 3.”

17. Results, line 368: please delete the word “altogether” (and the comma), as it is unnecessary. 

18. Results, line 370: This would be clearer if written as follows, “…whereas the most common approach in the post-guidance group was laparoscopic…” Please add “(Figure 3)” at the end of this sentence and delete the sentence at line 373, “The proportion of patients who underwent each surgical approach are reported in Figure 3.”

19. Results, line 387, please spell out HMB and PMB.

20. Results, lines 392-393: “Pre-guidance, we observed an 9.4% (8/85; 95% CI [4.5, 18%]) risk of occult malignancy among women with 393 NEH; in the post-guidance group this risk was lower at 3.5% (3/86; 95% CI [1.1, 10%]).” Suggest revising this sentence to “Pre-guidance, eight (9.4%; 95% CI [4.5, 18%]) of 85 women with NEH had occult malignancy, whereas in the post-guidance group, three (3.5% (86%; 95% CI [1.1, 10%]) of 86 women had occult malignancy (RR 0.38 (0.099-1.41); Table 3.” The next sentence should be revised similarly. 

21. Results, lines 399-400: Please move the sentence at lines 401-402 to the beginning of this paragraph: “Follow-up status at 2 years following diagnosis was available for 2,856 women (Table 4). Then the next sentence should be revised to read, “Two years had not yet lapsed for 413 women, meaning that they were ineligible for the two-year measures.”

22. Results, line 404: Please remove “(not shown in table)”

23. Results, line 408; Please remove “also” as the previous sentence referred to an increase, not a decrease.

24. Results, lines 411-413: Please rephrase. Eg, “The proportion of women with AEH who received the recommended 2 x 3-month follows-ups in the first instance was 13% (19/148; 95% CI [8.4, 19%]) pre-guidance and 19% (41/219; 95% CI [14, 413 26%]) post-guidance.”

25. Results, lines 413-414: Please remove the sentence “The proportions of women followed up to hysterectomy or regression over the first two years are shown in Figure 4.”

26. Results, lines 414-417: For consistency, please change sentence to the past tense and add a call-out to figure 4: “When we grouped women by two-year intervals for time of diagnosis, the proportion of women with AEH who underwent hysterectomy or who achieved histological regression without hysterectomy remained stable over time, whereas among women with NEH, the proportion who achieved regression increased and the proportion who undergo hysterectomy decreased.”

27. Discussion, line 432: Please remove the subheading “Principle findings” and all subsequent subheadings in the Discussion, which should be one continuous section. This includes the “Strengths and limitations,” “Implications…” and “Conclusions” headings.

28. Discussion, lines 476-477: Please rephrase for tense. Eg, “Approximately 30% of women with EH still underwent abdominal hysterectomy between 2016-2020.”

29. Discussion, line 478: The inclusion of the word “either” in this sentence makes it appear incomplete (one expects an ‘or’ to follow the ‘either’). Please revise as appropriate: “This may mean that either some hospitals may not be able to offer all women laparoscopic hysterectomy.” 

PRESS
